# Adequacy of Pain Treatment in Radiotherapy Departments: Results of a Multicenter Study on 2104 Patients (Arise)

**DOI:** 10.3390/cancers14194660

**Published:** 2022-09-25

**Authors:** Costanza M. Donati, Elena Nardi, Alice Zamagni, Giambattista Siepe, Filippo Mammini, Francesco Cellini, Alessia Di Rito, Maurizio Portaluri, Cristina De Tommaso, Anna Santacaterina, Consuelo Tamburella, Rossella Di Franco, Salvatore Parisi, Sabrina Cossa, Vincenzo Fusco, Antonella Bianculli, Pierpaolo Ziccarelli, Luigi Ziccarelli, Domenico Genovesi, Luciana Caravatta, Francesco Deodato, Gabriella Macchia, Francesco Fiorica, Giuseppe Napoli, Milly Buwenge, Romina Rossi, Marco Maltoni, Alessio G. Morganti

**Affiliations:** 1Radiation Oncology, IRCCS Azienda Ospedaliero-Universitaria di Bologna, 40138 Bologna, Italy; 2Radiation Oncology, Department of Experimental, Diagnostic and Specialty Medicine, Alma Mater Studiorum University of Bologna, 40138 Bologna, Italy; 3Medical Statistics, Department of Experimental, Diagnostic and Specialty Medicine, Alma Mater Studiorum University of Bologna, 40138 Bologna, Italy; 4Dipartimento di Diagnostica per Immagini Radioterapia Oncologica ed Ematologia, Fondazione Policlinico Universitario “A. Gemelli” IRCCS, UOC di Radioterapia Oncologica, 00168 Roma, Italy; 5Radioterapia Oncologica ed Ematologia, Dipartimento Universitario Diagnostica per Immagini, Università Cattolica del Sacro Cuore, Largo Agostino Gemelli 8, 00168 Roma, Italy; 6IRCCS Istituto Tumori “Giovanni Paolo II”, 70124 Bari, Italy; 7General Hospital “Perrino”, 72100 Brindisi, Italy; 8U.O. di Radioterapia AOOR PAPARDO PIEMONTE, 98121 Messina, Italy; 9S.C. di Radioterapia dell’Istituto Nazionale Tumori Pascale, 80131 Napoli, Italy; 10Radioterapia Opera di S. Pio da Pietralcina, Casa Sollievo della Sofferenza, 71013 San Giovanni Rotondo, Italy; 11IRCCS CROB, Rionero in Vulture, 85028 Potenza, Italy; 12U.O. Radioterapia Oncologica-S.O. Mariano Santo, 87100 Cosenza, Italy; 13Radioterapia Università degli Studi G. D’Annunzio, 66100 Chieti, Italy; 14Radiotherapy Unit, Gemelli Molise Hospital, Catholic University of Sacred Heart, 86100 Campobasso, Italy; 15U.O.C.di Radioterapia e Medicina Nucleare, Ospedale Mater Salutis di Legnago, 37045 Verona, Italy; 16IRCCS Istituto Romagnolo per lo Studio dei Tumori (IRST) “Dino Amadori”, 47014 Meldola, Italy; 17Medical Oncology Unit, IRCCS Azienda Ospedaliero-Universitaria di Bologna, 40138 Bologna, Italy; 18Department of Experimental, Diagnostic and Specialty Medicine, Alma Mater Studiorum University of Bologna, 40138 Bologna, Italy

**Keywords:** observational study, multicenter, radiotherapy, pain, pain management index

## Abstract

**Simple Summary:**

Cancer pain is often inadequately treated, as shown by several clinical studies. This problem has been confirmed in different clinical settings but the reasons for this phenomenon are unclear. Furthermore, little evidence is available on the adequacy of pharmacological pain management in patients undergoing radiotherapy. Moreover, studies investigating possible predictors of inadequate pain management reported contradictory results. Therefore, in this analysis, we evaluated a large population of cancer patients undergoing radiotherapy. We recorded, similarly to previous studies, a 45% rate of patients with inadequate analgesic therapy. Furthermore, evaluating the characteristics of patients with inadequate analgesic treatment, we noted that the subjects with better general conditions or better prognostic factors are those most frequently receiving inadequate drug therapy.

**Abstract:**

Aim: The frequent inadequacy of pain management in cancer patients is well known. Moreover, the quality of analgesic treatment in patients treated with radiotherapy (RT) has only been rarely assessed. In order to study the latter topic, we conducted a multicenter, observational and prospective study based on the Pain Management Index (PMI) in RT Italian departments. Methods: We collected data on age, gender, tumor site and stage, performance status, treatment aim, and pain (type: CP—cancer pain, NCP—non-cancer pain, MP—mixed pain; intensity: NRS: Numeric Rating Scale). Furthermore, we analyzed the impact on PMI on these parameters, and we defined a pain score with values from 0 (NRS: 0, no pain) to 3 (NRS: 7–10: intense pain) and an analgesic score from 0 (pain medication not taken) to 3 (strong opioids). By subtracting the pain score from the analgesic score, we obtained the PMI value, considering cases with values < 0 as inadequate analgesic prescriptions. The Ethics Committees of the participating centers approved the study (ARISE-1 study). Results: Two thousand one hundred four non-selected outpatients with cancer and aged 18 years or older were enrolled in 13 RT departments. RT had curative and palliative intent in 62.4% and 37.6% patients, respectively. Tumor stage was non-metastatic in 57.3% and metastatic in 42.7% of subjects, respectively. Pain affected 1417 patients (CP: 49.5%, NCP: 32.0%; MP: 18.5%). PMI was < 0 in 45.0% of patients with pain. At multivariable analysis, inadequate pain management was significantly correlated with curative RT aim, ECOG performance status = 1 (versus both ECOG-PS3 and ECOG- PS4), breast cancer, non-cancer pain, and Central and South Italy RT Departments (versus Northern Italy).Conclusions: Pain management was less adequate in patients with more favorable clinical condition and stage. Educational and organizational strategies are needed in RT departments to reduce the non-negligible percentage of patients with inadequate analgesic therapy.

## 1. Introduction

Pain, depression, and fatigue are common cancer symptoms. They have been identified by the National Cancer Institute as “priority symptoms” needing assessment [1]. Pain is one of the most frequent clinical symptoms in cancer patients, resulting from primary cancer progression, metastases, and treatment adverse effects. In fact, it has been estimated that up to 90% of patients can suffer from nociceptive and/or neuropathic pain during the course of tumor disease [2,3]. 

Moreover, pain is a multidimensional syndrome, severely worsening a patient’s quality of life (QoL) due to physical and emotional impact [4,5,6,7]. In fact, in cancer patients, lack of pain control is the best predictor of worse QoL as a result of its negative effect on daily activities, mood, and personal independence [8,9]. 

Therefore, pain relief represents a priority in oncology, and pain evaluation before and during treatment is recommended to treat this symptom effectively [3,10]. Unfortunately, inadequate treatment of pain is frequent despite the availability of guidelines for cancer pain management and of several effective analgesic therapies [11,12,13,14,15,16]. 

For this reason, many studies evaluated pain management in different cancer settings [17,18,19,20,21,22,23,24,25,26,27,28]. However, only a few reports on this topic are available for patients treated with radiotherapy (RT). Therefore, we planned a multicenter observational study to assess the adequacy of pain management in cancer patients treated with RT in Italian centers. 

## 2. Patients and Methods 

### 2.1. Study Aims

The primary objective of the trial was to evaluate the adequacy of pain management in patients treated in RT departments. The secondary objective was to evaluate any correlation between adequacy of pain management and potential predictors (gender, age, performance status, timing of the visit, RT aim, primary tumor, stage of disease, type of pain and geographical location of the RT center).

### 2.2. Study Design 

It was an observational, prospective, multicenter cohort study. Patients were enrolled after signing the informed consent. The study was approved by the Ethics Committees of participating centers (ARISE 327/2017/O/Oss). All patients who underwent a medical examination in the participating centers were considered for the study enrollment. All patients who met the enrollment criteria and who underwent a clinical visit at least once in the RT departments of participating centers in the period October–November 2019 were included. The evaluation was performed regardless of the visit timing (ongoing RT visits or clinical evaluation at the end of treatment). However, each patient was evaluated only once. The data were recorded through a collection form filled in during the visit. Data on gender, age, Eastern Cooperative Oncology Group Performance Status Scale (ECOG-PS), RT aim, primary cancer, tumor stage, intensity of pain measured with the Numeric Rating Scale (NRS), analgesic score and type of pain (cancer pain: CP, non-cancer pain: NCP, mixed pain: MP) were collected. 

### 2.3. Inclusion Criteria

Inclusion criteria were: (1) cancer patients (regardless of stage, primary tumor, tumor stage, and RT aim), (2) treated in RT departments, (3) aged ≥ 18 years. Patients with comorbidities (psychiatric disorders or neurosensory deficits) preventing data collection or granting of consent were excluded.

### 2.4. End Points

We assigned a pain score by using the following values: 0 (NRS: 0, no pain), 1 (NRS: 1–4, mild pain), 2 (NRS: 5–6, moderate pain), and 3 (NRS: 7–0, intense pain). In addition, based on the therapy the patients took, we defined an analgesic score as follows: no analgesics: 0, non-opioid analgesics: 1, “weak”opioids: 2, and “strong” opioids: 3. The Pain Management Index (PMI) was calculated by subtracting the pain score from the analgesic score, considering prescriptions with a negative value as inadequate [29,30]. 

### 2.5. Statistical Analysis

Gender, age, PS, timing of the visit, RT aim, primary tumor, stage of disease, type of pain, analgesics score, and RT center were explored as potential correlations with PMI. Using SYSTAT (version 11.0, SPSS, Chicago, IL, USA) we evaluated the correlation between PMI and potential predictors with the chi-squared test, considering values < 0.05 as significant. Furthermore, we included in the multivariate analysis (multiple logistic regression) the variables found to be statistically significant at the univariate analysis, in order to confirm the predictive impact of potential predictors of inadequate PMI.

## 3. Results

### 3.1. Patient Characteristics

Overall, 2104 patients were enrolled in the study, of which 1417 complained of pain and 1090 were taking analgesic drugs. Patient characteristics are shown in Table 1.

### 3.2. Pain Management Index (PMI)

Considering all patients enrolled in the study, the rate of subjects with PMI < 0 was 30% (Figure 1). Furthermore, concerning only patients with pain or receiving analgesics, the PMI value was <0 in 639 subjects (45.0%) (Figure 2). Of patients enrolled and undergoing palliative and curative RT, 28% (Figure 3) and 32% (Figure 4) showed PMI < 0, respectively. Instead, considering only patients with pain of the subjects undergoing palliative and curative RT, 30% (Figure 5) and 62% (Figure 6) showed PMI < 0, respectively.

### 3.3. Predictors of Pain Management Adequacy

At univariate analysis, performed only on patients with pain or taking analgesics, the following parameters were significantly correlated with PMI < 0: female gender, curative treatment aim, lower ECOG-PS score, breast cancer, non-cancer pain, non-metastatic stage, RT department in the center or south of Italy (Table 2). The multivariate analysis, in the same patient population, confirmed the significant correlation with PMI < 0 of the following parameters: ECOG-PS1 (versus both ECOG-PS3 and ECOG-PS4), breast cancer (versus prostate, gastrointestinal, uterine, head and neck, and other cancers), non-cancer pain (versus cancer-related pain), and location of the RT center in the center or south of Italy (versus northern Italy) (Table 3). 

## 4. Discussion

In a multicenter study including over two thousand patients evaluated during RT, the rate of patients with inadequate pain management (PMI < 0) was 45.0%. The inadequacy of analgesic therapy was significantly correlated to different parameters: (i) the patient’s physical condition (more frequent in subjects with better performance status), (ii) the type of tumor treated (more frequent in breast cancer), (iii) the origin of pain (more frequent in non-neoplastic pain), and (iv) the geographic location of the RT department (more frequent in central and southern Italy). 

Finally, the lack of pathophysiological classification of pain (nociceptive versus neuropathic versus mixed), in our analysis, did not allow the evaluation of the impact of this parameter on the adequacy of pain management.

Some observations can be made by comparing our results with some previous analyses (Table 4). PMI is more frequently negative in patients undergoing curative treatment than in those undergoing palliative RT. A similar result was previously reported by Fujii et al., who observed a significantly higher rate of patients with PMI < 0 in subjects undergoing adjuvant chemotherapy compared to patients receiving chemotherapy for advanced disease [24]. Furthermore, in our analysis a PMI < 0 was more frequently observed in patients with ECOG-PS 1 than in ECOG-PS 3–4. This result confirms other studies reporting similar correlations [17,24]. Moreover, both the association with a palliative aim of RT and that with worse ECOG-PS suggests greater attention to pain management in patients in worse clinical conditions.

Furthermore, a negative PMI is more common in breast cancer patients than in all other cancers (60.5% versus 30.9–49.4%). These data cannot be explained simply and in particular, at least in our series, cannot be interpreted just on the basis of the female gender. In fact, our multivariate analysis did not show a significant correlation between gender and PMI. Furthermore, a significantly higher rate of PMI < 0 was recorded also compared to other female cancers (endometrium and uterine cervix). Moreover, the correlation between PMI < 0 and breast cancer was also reported in other studies [17,25,28]. We could reasonably hypothesize that breast cancer patients have several factors predisposing them to poor pain management. In fact, in most cases, they are patients undergoing adjuvant RT after surgery and therefore: (i) they are in good clinical conditions (ECOG-PS 1), (ii) if they suffer from pain this often depends on previous surgery (and therefore the origin of the symptom is non-neoplastic), (iii) they receive an adjuvant treatment (and therefore with curative purposes). 

However, these explanations are not convincing given that the same correlation between PMI < 0 and breast cancer was recorded in a study including only metastatic patients undergoing palliative RT [17]. Other authors tried to interpret this finding otherwise and in particular considering the greater sensitivity to pain of female patients [31], their lower compliance with analgesic intake and a tendency to stop therapy early, at the first signs of improvement [14], and, more generally, the complexity of pain management in breast cancer, as this symptom is part of clusters that also include fear of relapse, fatigue, and anxiety [32]. Finally, another possible explanation could concern the high incidence of bone metastases, often painful, in this patient population. However, the lack of registration of the sites of metastatic disease, in our database, precludes a confirmation analysis of this hypothesis.

Furthermore, PMI < 0 is more frequent in case of non-neoplastic pain, i.e., produced by benign comorbidities. This result is similar to the findings of two previously published analyses [19,25]. Finally, negative PMI values are more common in patients treated in southern and central Italy. Geographic variations in the adequacy of analgesic therapy within the same country were previously reported in a study conducted in Taiwan [25].

According to a literature review, PMI < 0 is recorded in about 43% of cancer patients [15], although a trend towards a reduction of this figure has been observed in recent decades [33]. Our result (PMI < 0:45.0%) is similar to that reported in the cited literature review [15] and in other similar PMI-based analyses (PMI < 0:39.7–53.0%) [24,27,28]. Instead, other analyses recorded worse results (PMI <0:77–83%) [19,22]. One study was on patients treated over 10 years ago in a center of southern Italy [19], and the other an analysis on a particularly young population (≤60 years: 75% of patients) and therefore probably in relatively good clinical condition, a status that in our and other analyses correlates with higher negative PMI rates [22]. Conversely, other analyses recorded lower rates (4–33.3%) of inadequate pain management [17,18,20,21,23,25,26]. In some cases [17,18,21] this result can be explained by the enrollment of patients undergoing only palliative treatment, which, from ours and Fujii’s et al. [24] analyses, correlates with better pain management Instead, in other cases, the improved adequacy of pain treatments may be due to patient management in supportive or palliative care departments [20,23,26].

This analysis has several limitations. In fact, the study analyzed only the pain management but not the impact on quality of life. Furthermore, the PMI assessment was performed at different times (during or at the end of RT), with only one evaluation per patient. Therefore, it is difficult to assess how much inadequate pain management is attributable to the physicians who treated the patients prior to RT or to radiation oncologists. However, in patients assessed at the end of RT, the rate of negative PMI was lower compared to patients evaluated during treatment (38.0% versus 47.0%; *p*: 0.015). Both the progressive adjustment of drug therapy during RT and the analgesic effect of RT in patients undergoing palliative treatment could have led to this difference. 

Another weak point of the study is the known limitations of the PMI, the tool we used to assess the adequacy of pain management. In fact, the PMI is based on the obsolete distinction between weak and strong opioids [30]. Furthermore, the correlation of PMI with quality of life is questionable. Indeed, a PMI < 0 is not significantly correlated with patients’ desire to receive greater attention to their pain [28]. However, lower PMI values are generally correlated with a higher percentage of patients complaining of pain interfering with their daily life [26]. 

Another limitation of the PMI is that this index is generally calculated on the basis of the analgesic therapy prescribed and not of that actually taken by the patients [29,30,34,35]. However, our study was conducted by interviewing patients and then gathering information on the therapy taken and not on the prescribed ones. Nevertheless, the PMI’s main limitation is to consider all patients taking strong opioids as adequately treated. In fact, all these subjects have a PMI value ≥ 0, regardless of the type and dose of the drugs, and especially of the degree of pain relief [30]. However, given its correlation with the quality of pain treatment and the easy calculation and collection, the PMI remains the most frequently used surrogate indicator of the appropriateness of pharmacological pain management [36]. 

## 5. Conclusions

In conclusion, the result of our and other analyses suggest that the attention to adequate pain therapy is lower in patients with better clinical conditions (good PS, non-neoplastic pain) and with a more favorable prognosis (RT for curative purposes). 

Moreover, the near to 50% rate of patients not receiving adequate analgesic therapy in RT departments deserves attention. Therefore, in this clinical setting, it could be useful: (i) to implement the systematic registration of PMI, in addition to that of pain, to screen patients with inadequate pain management, (ii) to promote educational strategies for medical and nursing staff aimed at improving the awareness of this topic and the ability to adequately identify and treat patients with painful symptoms, (iii) to improve symptom management also through multidisciplinary collaborations (multidisciplinary teams, joint clinics).

Furthermore, considering several points not fully clarified by the published reports, further research seems necessary. Future studies could have the following aims: (i) to prospectively analyze the evolution of pain and its management during the path of patients in RT departments, to identify opportunities for optimization, (ii) to test the impact of educational strategies aimed at improving knowledge and skills of radiation oncologists in non-invasive pharmacological pain management, and (iii) to analyze the characteristics of pain in patients referred to RT to possibly optimize timing and methods of radiation oncologists’ consultations.

## Figures and Tables

**Figure 1 cancers-14-04660-f001:**
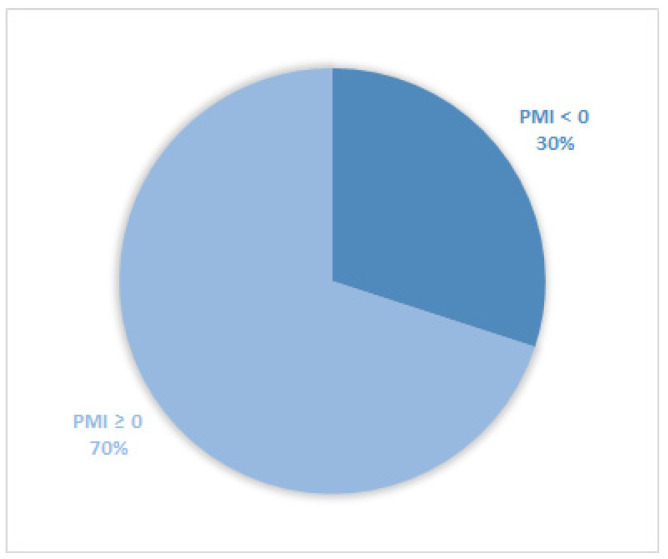
Pie chart displaying the percentage of patients with PMI < 0 and PMI ≥ 0. All patients were included (2104).

**Figure 2 cancers-14-04660-f002:**
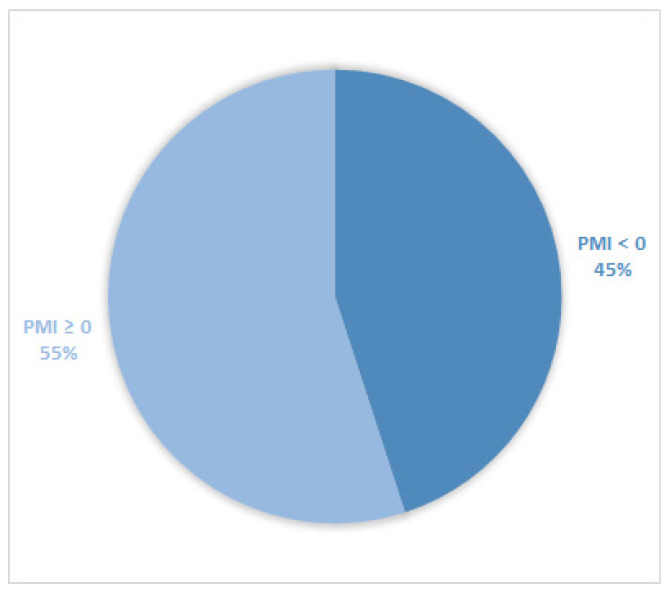
Pie chart displaying the percentage of patients with PMI < 0 and PMI ≥ 0. Only patients with pain or receiving analgesics were included (1417).

**Figure 3 cancers-14-04660-f003:**
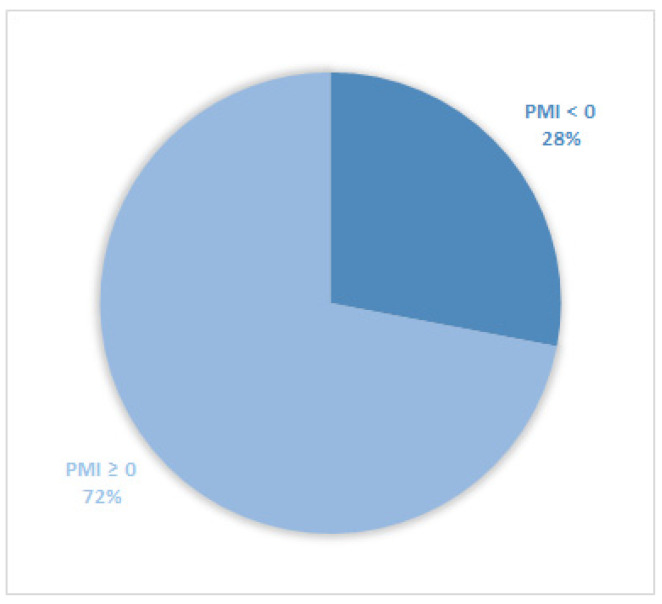
Pie chart displaying the percentage of patients with PMI < 0 and PMI ≥ 0. Only patients undergoing palliative radiotherapy were included (791).

**Figure 4 cancers-14-04660-f004:**
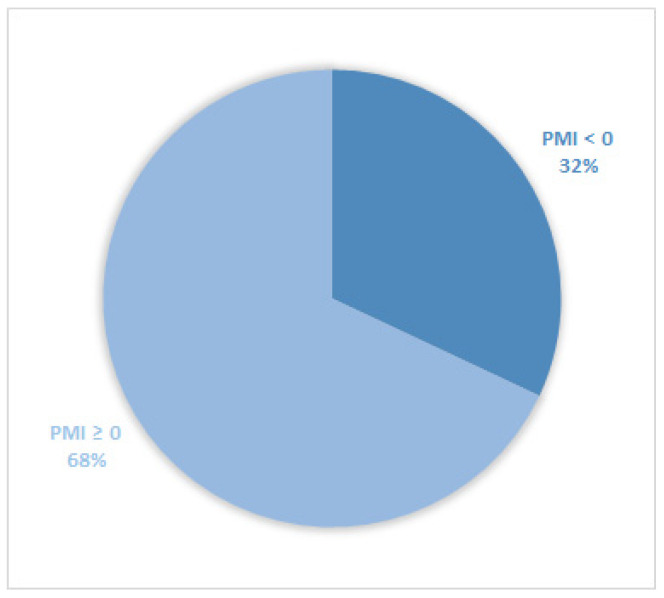
Pie chart displaying the percentage of patients with PMI < 0 and PMI ≥ 0. Only patients undergoing curative radiotherapy were included (1313).

**Figure 5 cancers-14-04660-f005:**
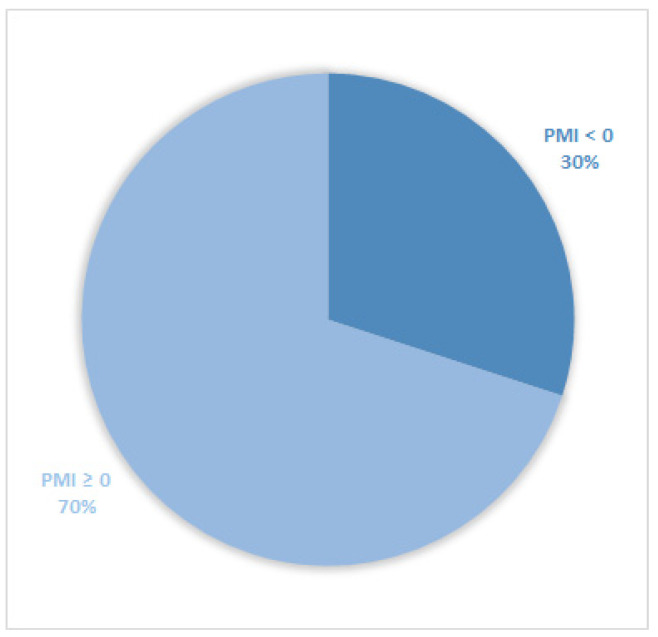
Pie chart displaying the percentage of patients with PMI < 0 and PMI ≥ 0. Only patients undergoing palliative radiotherapy and with pain or receiving analgesics were included (737).

**Figure 6 cancers-14-04660-f006:**
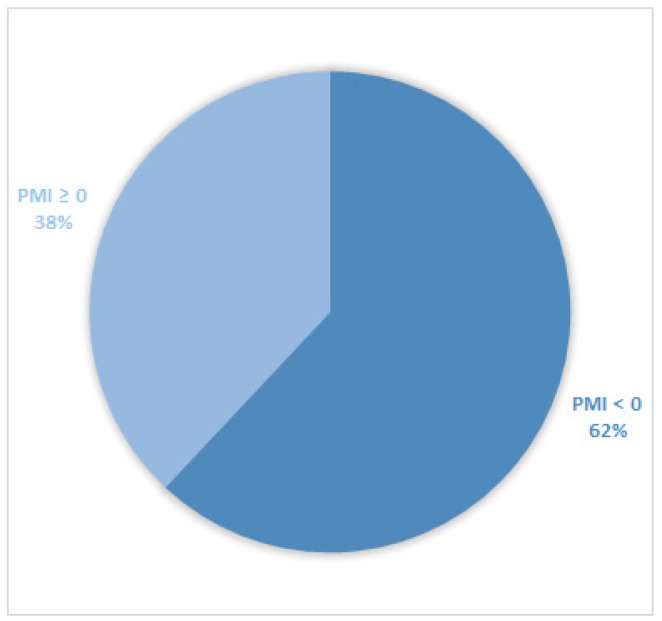
Pie chart displaying the percentage of patients with PMI < 0 and PMI ≥ 0. Only patients undergoing curative radiotherapy and with pain or receiving analgesics were included (680).

**Table 1 cancers-14-04660-t001:** Patient characteristics.

Patient Characteristics		Number	(%)
** *Gender* **			
Male		951	45.2
Female		1153	54.8
** *Age, years* **			
≤70		1340	63.7
71–80		577	27.4
>80		187	8.9
** *ECOG-PS* **			
0		582	27.7
1		963	45.8
2		358	17.0
3		171	8.1
4		30	1.4
** *Aim of treatment* **			
Curative		1313	62.4
Palliative		791	37.6
** *Primary Tumor* **			
Breast		695	33.0
Prostate		302	14.4
Gastrointestinal		207	9.8
Endometrial/Cervical		143	6.8
Lung		235	11.2
Head and Neck		159	7.6
Others		363	17.2
** *Tumor stage* **			
Metastatic		899	42.7
Non-metastatic		1205	57.3
** *Type of Pain* **			
Cancer Pain		701	49.5
Non-cancer Pain		456	32.0
Mixed Pain		260	18.5
** *Pain score* **			
(NRS: 0)	0	751	35.7
(NRS: 1–4)	1	591	28.1
(NRS: 5–6)	2	509	24.2
(NRS: 7–10)	3	253	12.0
** *Analgesic score* **			
(No therapy)	0	1014	48.0
(Analgesics)	1	592	28.0
(Weak Opioids)	2	202	10.0
(Strong Opioids)	3	296	14.0
** *Location of the radiotherapy center* **			
Nord of Italy		484	23.0
Center of Italy		349	16.6
South of Italy		1271	60.4
** *Timing of visit* **			
During Therapy		1770	84.1
End of Therapy		334	15.9

Legend: ECOG-PS: Eastern Cooperative Oncology Group Performance Status Scale; NRS: Numeric Rating Scale.

**Table 2 cancers-14-04660-t002:** Univariate analysis on Pain Management Index (only 1417 patients with pain or under analgesic therapy included).

				PMI				*p*-Value
		All Patients		<0		≥0		
				*n*	%	*n*	%	
All Patients				639	45.0	778	55.0	
**Gender**	Male	693	49.0	264	38.0	429	62.0	<0.001
	Female	724	51.0	375	52.0	349	48.0	
**Age, years**	≤70	879	62.0	373	42.4	506	57.6	0.008
	71–80	380	27.0	197	52.0	183	48.2	
	>80	158	11.0	69	43.7	89	56.3	
**Aim of treatment**	Curative	680	48.0	421	62.0	259	38.0	<0.001
	Palliative	737	52.0	218	29.6	519	70.4	
**ECOG-PS**	0	30	2.0	0	0.0	30	100.0	
	1	852	60.0	468	55.0	384	45.0	
	2	343	24.5	122	35.6	221	64.4	<0.001
	3	163	11.5	45	27.6	118	72.4	
	4	29	2.0	4	13.8	25	86.2	
**Primary Tumor**	Breast	435	30.5	263	60.5	172	39.5	
	Prostate	149	10.5	65	43.6	84	56.4	<0.001
	Gastrointestinal	139	9.8	43	30.9	96	69.1	
	Endometrial/Cervical	79	5.6	39	49.4	40	50.6	
	Lung	202	14.3	58	28.7	144	71.3	
	Head and Neck	126	9.0	62	49.2	64	50.8	
	Others	287	20.3	109	38.0	178	62.0	
**Type of Pain**	Cancer Pain	701	49.5	214	30.5	487	69.5	<0.001
	Non-cancer Pain	456	32.2	326	71.5	130	28.5	
	Mixed Pain	260	18.3	99	38.1	161	61.9	
**Tumor stage**	Non-metastatic	649	45.8	399	62.0	250	38.0	<0.001
	Metastatic	768	54.2	240	31.2	528	68.8	
**Location of the radiotherapy center**	North of Italy	291	20.5	103	35.4	188	64.6	<0.001
	Center of Italy	177	12.5	102	57.6	75	42.4	
	South of Italy	949	67.0	434	45.7	515	54.3	
**Timing of visit**	During therapy	1175	83.0	547	47.0	628	53.0	0.015
	End of therapy	242	17.0	92	38.0	150	62.0	

Legend: ECOG-PS: Eastern Cooperative Oncology Group Performance Status Scale; NRS: Numeric Rating Scale. Percentages in “all patients” columns are column percentages. Percentages in “PMI” columns are row percentages.

**Table 3 cancers-14-04660-t003:** Multivariable analysis (only 1417 patients with pain or under analgesic therapy included).

Patient Characteristics	OR	S.E.	95% CI	*p*-Value
**Gender**				
Male	-			
Female	1.089	0.143	0.899–1.492	0.147
**Aim of treatment**				
Curative	-			0.000
Palliative	0.437	0.168	0.314–0.607	
**ECOG-PS**				0.126
1	-			
2	0.787	0.157	0.579–1.070	
1	-			0.012
3	0.584	0.213	0.385–0.887	
1	-			0.023
4	0.277	0.564	0.092–0.838	
**Primary Tumor**				0.002
Breast	-			
Prostate	0.500	0.224	0.322–0.776	
Breast	-			0.000
Gastrointestinal	0.346	0.231	0.220–0.545	
Breast	-			0.032
Endometrial/Cervical	0.546	0.282	0.315–0.949	
Breast	-			0.000
Lung	0.481	0.207	0.321–0.720	
Breast	-			0.001
Head and Neck	0.473	0.234	0.299–0.749	
Breast	-			0.030
Others	0.681	0.177	0.481–0.963	
**Type of pain**				0.000
Cancer Pain	-			
Non-cancer Pain	2.630	0.172	1.879–3.683	
Cancer pain	-			0.380
Mixed Pain	1.152	0.161	0.840–1.580	
**Location of the radiotherapy center**				0.001
Nord of Italy	-			
Center of Italy	2.179	0.224	1.404–3.381	
Nord of Italy	-			0.001
South of Italy	1.747	0.163	1.270–2.404	

Legend: ECOG-PS: Eastern Cooperative Oncology Group Performance Status Scale; OR: odds ratio; 95% CI: 95% confidence interval.

**Table 4 cancers-14-04660-t004:** Comparison with other studies on Pain Management Index evaluated in cancer patients.

Author	Center	No. of Patients (Patients with Pain/Total)	Setting and Methods	Results
Mitera G., 2010 [17]	Odette Cancer Centre, Sunnybrook Health Sciences Centre, University of Toronto, Canada	981/1000	Retrospective analysis of PMI in initial assessment or in follow-up in pts with bone metastases enrolled in a Rapid Response Radiotherapy Program	PMI < 0:25.3% (initial consultation); 15.4–17.5% (follow-up) * PMI < 0 correlated with better PS and breast cancer
Mitera G., 2010 [18]	Odette Cancer Centre, Sunnybrook Health Sciences Centre, University of Toronto, Canada	2011	Prospective and multicenter analysis of PMI in pts with bone metastases treated in a palliative radiotherapy clinic	PMI < 0:25.1% *; moderate to severe pain: 70.9%
Massaccesi M., 2013 [19]	Università Cattolica del S. Cuore, Campobasso, Italy	398/865	Prospective analysis of PMI in initial assessment or in follow-up in cancer pts (radiation oncology unit)	PMI< 0: 82.6%; NCP > CP; NCP 91.4%
Gonçalves F., 2012 [20]	Instituto Português de Oncologia, Porto, Portugal	136/164	Ten palliative care teams participate in a prospective cross-sectional survey of PMI in subjects (mainly neoplastic: 92%) hospitalized or outpatient or followed at home by a hospital team	PMI < 0:4%
Vuong S., 2016 [21]	Odette Cancer Centre, Sunnybrook Health Sciences Centre, University of Toronto, Canada	354	Retrospective analysis of PMI in pts with bone metastases treated within a Rapid Response Radiotherapy Program in a palliative radiotherapy clinic	PMI < 0:33.3% *
Singh H., 2017 [22]	Baba Farid University of Health Sciences, Faridkot, India	348/348	Observational prospective analysis of PMI and BPI in pts admitted to an oncology department	PMI < 0:77%
Reis-Pina P., 2017 [23]	Pain Clinic, of the Portuguese Cancer Institute, Lisbon, Portugal	371/371	Prospective analysis of PMI in cancer pts during the first consultation in an outpatient pain clinic	PMI < 0:25.6%; PMI < 0 correlated with: female gender, recent RT treatment, neuropathic pain, adjuvant analgesics
Fujii A., 2017 [24]	Research Institute for Diseases of the Chest, Kyushu University, Fukuoka, Japan	365/524 and 320/524	Observational longitudinal study of PMI in initial assessment or in follow-up of outpatients treated in a chemotherapy unit	PMI < 0:39.7% (initial consultation); PMI < 0:51.7% (follow-up); PMI < 0 correlated with better PS, adjuvant chemotherapy, depressive state
Shen W.C., 2017 [25]	Division of Hematology-Oncology, Linkuo Chang Gung Memorial Hospital and Chang Gung University, Taoyuan, Taiwan	1659/2652	Observational prospective analysis of PMI in outpatients treated in 16 centers (oncologic clinics)	PMI < 0:32.4%; PMI < 0 correlated with female gender, breast cancer, NCP, north Taiwan hospital
Sakakibara N., 2018 [26]	Department of Palliative Care, St. Luke’s International Hospital, Tokyo, Japan	1156 (3682/6732 responses)	Prospective observational study on PI (pain interference) across various PMI scores in hospitalized cancer patients	PMI < 0:26.6% PMI -3/-2 correlated with PI of 72.3% and 63.3% respectively
Tuem K. B., 2020 [27]	Department of Pharmacology and Toxicology, Mekelle University, Mekelle, Ethiopia	91/91	Observational prospective analysis of PMI and BPI in pts admitted to an oncology department	PMI < 0:43.9%
Thronæs M., 2020 [28]	Cancer Clinic, St. Olavs Hospital, Trondheim University Hospital, Trondheim, Norway	187/187	Observational prospective analysis of PMI in pts admitted in departments of oncology, surgery, internal medicine, and gynecology	PMI < 0:53%; PMI < 0 correlated with KPS > 50%, breast cancer, and evaluation during follow-up
*Present series*, 2022	Radiation Oncology, Bologna University, Bologna, Italy	1409/2104	Observational prospective analysis of PMI in pts treated in 13 radiation oncology departments	PMI < 0:45.4%

Legend: BPI: Brief Pain Inventory; CP: cancer-related pain; NCP: non-cancer related pain; PI: pain interference; PMI; Pain Management Index. * PMI calculated on all patients.

## Data Availability

Data supporting the reported results will be made available on reasonable request.

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
