# Peer review of "Adequacy of Pain Treatment in Radiotherapy Departments: Results of a Multicenter Study on 2104 Patients (Arise)"

_cancers, 2022, doi:10.3390/cancers14194660_

Round 1

Reviewer 1 Report

The manuscript addresses the important clinical issue of not adequate care for pain, especially in people with cancer undergoing treatment with curative intent. 

The text is well written and clearly presents the aim, material and methods, results and discussion.

I would suggest only small corrections / adjustments:

1. Row 150 the word "Furthermore" should be corrected (currently spelling in manuscript  "further-more").

2. Title of Table 3 is too general, should be more detailed "Multivariate analysis" on .... I am wondering if a graphic presentation, instead of the numbers would not be more convincing and easier for readers to overlook the results. I am missing presenting the influence of gender on PMI<0 OR (described in Row 250). Even if statistically insignificant, it could be presented, as this issue is analysed in the discussion.  If the authors will decide not to follow this advice - removing the " [ref]" would make the text more readable. In the text (row 243) the risk of PMI<0 in the subpopulation with ECOG1 has been compared with the subpopulation with ECOG 3-4. In the table, those data are presented separately for ECOG 3 and ECOG 4. I would suggest a more homogenous presentation of the data.  

One general comment to the wording - the term "gender" is not completely wrong (however per definition it refers to the sex identity), but maybe replacing it with "sex category" would be more appropriate?  

Reviewer 2 Report

Dear Authors,

it is interestng study based on a large number of patients. However, I would suggest some points that you may wish to consider:

1. Your concept of Modified Pain Management Index (MIAMI) is not clear and may not reflect appropriate pain management. This is associated with some limitations (that you discussed in detail) of an original Pain Management Index. Therefore, building another concept based on that seems to be risky.

Moreover, an assumption that all patients with PMI < 0 or patients with PMI ≥ 0 but with pain score > 1 (NRS > 4) were considered to have a negative MIAMI scores may be not true and as you stated in the Discussion section this approach this concept was not formally validated.

Therefore, I would suggest to remove the MIAMI concept from the manuscript.

2. I would suggest to re-write the Discussion section with placing limitations at the end of the chapter before Conclusions and starting the Discussion from presenting your results, then comparing it with other studies conducted. The last part devoted to future studies and recommendations for clinical practice may be more concise.

3. I would suggest changing the title of Section 2 to "Patients and methods" instead of "Material and methods".

4. One of possible explanations for breast cancer patients poorer pain management may be high frequency of bone metastases and in consequence bone pain in these patients.

5. I would mention lack of type of pain assessment regarging pathophysiology: nociceptive, neuropathic, and mixed (including bone) as one of important limitations of the study

Round 2

Reviewer 2 Report

Dear Authors,

thank you for your responses. However, I cannot agree with your concept of MIAMI. The assumption that patients with PMI < 0, which is an indicator of inappropriate pain treatment are the same as patients with PMI = 0 or PMI > 0 (which are both appropriate pain treatment indicators according to PMI) and pain intensity according to NRS > 4 as an indicator of poor pain management is not the case.

As your MIAMI concept was not validated, I would suggest to remove it from this manuscript, which will improve its clarity.

You may indicate in the Discussion section that PMI concept has its limitations.

Otherwise I have no critical comments

Round 3

Reviewer 2 Report

Dear Authors,

thank you for all amendments. My only suggestion is to correct last part of Results section in an Abstract regarding inadequate pain treatment in locations of radiotherapy centers in Central and South Italy vs. Northern Italy (multivariable analysis) to be in agreement with the text of Results and Discussion sections in your manuscript.

I congratulate you on publication of this important manuscript